# Phenolic Profiling and In-Vitro Bioactivities of Corn (*Zea mays* L.) Tassel Extracts by Combining Enzyme-Assisted Extraction

**DOI:** 10.3390/foods11142145

**Published:** 2022-07-20

**Authors:** Nesren Elsayed, Diaa A. Marrez, Mohamed A. Ali, Ahmed Ali Abd El-Maksoud, Weiwei Cheng, Tarek Gamal Abedelmaksoud

**Affiliations:** 1Food Science Department, Faculty of Agriculture, Cairo University, Giza 12613, Egypt; nesrensayed@agr.cu.edu.eg; 2Food Toxicology and Contaminants Department, National Research Centre, Cairo 12622, Egypt; diaamm80@hotmail.com; 3Biochemistry Department, Faculty of Agriculture, Cairo University, Giza 12613, Egypt; mohamed_soliman@cu.edu.eg; 4Department of Dairy Science, Faculty of Agriculture, Cairo University, Giza 12613, Egypt; ahmed_ali@cu.edu.eg; 5Institute for Innovative Development of Food Industry, Institute for Advanced Study, Shenzhen University, Shenzhen 518060, China

**Keywords:** corn tassel, phenolic profile, enzyme-assisted extraction, antioxidant, antimicrobial, cytotoxicity

## Abstract

In this work, enzyme-assisted extraction (EAE) of phenolic compounds from corn tassel using cellulase, protease, and their combination (1:1) was developed and optimized by central composite response surface methodology. The phenolic profile of obtained corn tassel extracts (CTE) was elucidated by high-performance liquid chromatography–diode array detection (HPLC–DAD) analysis, and their antioxidative, antimicrobial, and cytotoxic properties were evaluated in vitro. The results showed that CTE by EAE with combined enzymes had the highest total phenolic content (TPC). Under optimum enzymatic conditions, the experimental TPC values were 9.78, 8.45, and 10.70 mg/g, respectively, which were significantly higher than that of the non-enzymatic control (6.75 mg/g) (*p* < 0.05). Fourteen more phenolic compounds (13.80–1694.36 µg/g) were identified in CTE by EAE with the combined enzymes, and thus the antioxidant activity of that extract, determined by DPPH and ABTS radical scavenging method, was demonstrated to be stronger than that of the extracts by EAE with the single and ethanol extraction. Furthermore, this extract also showed remarkably better antimicrobial properties against all tested food-borne pathogenic bacteria and mycotoxigenic fungi than CTE by other extraction methods. CTE by EAE were nontoxic to normal lung fibroblast cell line (Wi-38) but cytotoxic to human colorectal and lung cancer cell lines (Caco-2 and A549), with IC_50_ values of 392.62–461.98 and 210.66–359.56 µg/mL, respectively, which indicated its potential anticancer properties. In conclusion, CTE by EAE, especially with the combined use of cellulase and protease, seems to hold promising potential for multifunctional application in food and pharma fields.

## 1. Introduction

Corn (*Zea mays* L.) is one of the most extensively grown crops in the world, with an estimated global yield of 13.7 billion bushels [1]. Corn production in Egypt was 8 tons per ha [2]. Corn tassel is a waste part of the plant, and the chemical composition consists of carbohydrates 70.26%, ash 11.06%, fat 10.10%, and protein 6.26% [3]. In addition, corn tassel is considered an important source of valuable products such as volatile oils, lipids [4], phenolic compounds, and flavonoids such as quercetin, isorhamnetin, and kaempferol [5]. The main bioactive compounds in corn tassel are phenolic substances, which possess a variety of biological and physiological properties, including antioxidative, antibacterial, anticancer, anti-allergic, anti-atherogenic, anti-inflammatory, anti-thrombotic, cardioprotective, and vasodilatory actions [6]. To our knowledge, the most common method for extracting phenolic compounds is conventional solvent extraction. However, the disadvantage of solvent extraction is the loss of the phenolic compounds due to long extraction periods and some phenolic compounds’ degradation. In addition, conventional extraction requires a large amount of solvent, which is not economical. According to Pinelo et al. [7], various parameters influence extraction efficiency, including the type of solvent, solvent extraction, extraction time, extraction temperature, and solid-solvent ratio.

Recently, based on the avoidance of chemical ingredients’ side effects in foodstuffs, researchers have tried to replace conventional solvent extraction with green methods for the extraction of phytochemicals from plant sources such as enzymes, ultrasound, high hydrostatic pressure, etc. Enzymatic-assisted extraction as one of the green methods was applied to rupture the plant cell wall polysaccharides to enhance the release of bioactive compounds intermingled in plants.

Phenolic compounds in plant cell walls are both chemically diverse and botanically widespread. Therefore, in theory, the enzymatic treatment may extract more phenolic chemicals than solvent extraction. Many studies have also well used enzymes in the extraction of phenolic compounds [8,9]. However, until now, there are no reports available for using the enzymes to extract phenolic compounds from corn tassel. Furthermore, there is little information available concerning the bioactivities of corn tassel extract.

Therefore, the present work aims to improve the enzyme-assisted extraction process of phenolic components from corn tassels and assess the antioxidative, antimicrobial, and cytotoxic properties of extracts. To this purpose, a standard 2-factor central composite response surface design was adopted to optimize the phenolic recovery from corn tassel by enzyme-assisted extraction. The phenolic composition in corn tassel extract was determined by high-performance liquid chromatography–diode array detection (HPLC–DAD). Furthermore, the scavenging activities against DPPH and ABTS radicals, antimicrobial activities against food-borne pathogenic bacteria and mycotoxigenic fungi, and cytotoxic effects against cancer and normal cell lines were performed in our work. The results gathered in this paper would provide a solid foundation for the development of new value-added products from corn tassel, such as soft drinks and juices.

## 2. Materials and Methods

### 2.1. Chemicals

All reagents and chemicals used in this research were of analytical grade. Gallic acid, Folin-Ciocalteu reagent, ABTS 2,2′-azinobis (3-ethylbenzothiazoline-6-sulphonic acid), 2,2-diphenyl-1-picrylhydrazyl (DPPH), 3-(4,5-Dimethylthiazol-2-yl)-2,5-Diphenyltetrazolium Bromide (MTT), ethanol, sodium bicarbonate, sodium acetate, and butylated hydroxytoluene (BHT) were obtained from Sigma Chemical Co., Ltd. (St. Louis, MO, USA). Cellulase (Novozymes, Mumbai, India) (3500 U/g non-GMO) and Proteases (VemoZyme, Mumbai, India) (15,000 U/g) enzymes were obtained from Alfa-chemicals, El-Dokki-Giza, Egypt. Tryptic soy broth and Potato Dextrose agar were obtained from Conda Laboratories (Madrid, Spain).

### 2.2. Materials

Corn tassels were collected from the field of the Faculty of Agriculture, Cairo University, Egypt. Corn tassels were dried in an oven (Shel-lab, Cornelius, OR, USA) at 40 °C for 48 h until no further weight reduction was measurable, then subjected to grinding by an analytical mill to a size of 1 mm (Cole-Parmer, Vernon Hills, IL, USA), sieved up to 50 mesh and kept in the dark place at room temperature for analysis.

### 2.3. Conventional Extraction

The ethanolic solvent for the extraction of phenolic compounds was used. As described by Younis et al. [10], with some modifications, about one gram of dried corn tassel powder was added to 20 mL 60% (*v*/*v*) ethanol–water solutions and then subjected to a shaking incubator at room temperature for 24 h, followed by filtration through Whatman No. 4 filter paper to get pure extracts. The extracts were kept at −18 °C until further analysis.

### 2.4. Experimental Design and Statistical Analysis

Cellulase and protease were used either individually or their binary mixture (1:1) to extract phenolic compounds from corn tassels. The concentration of each enzyme (cellulase and protease) for extracting phenolic compounds was studied. Experimental variables were optimized to maximize the yield of extracted polyphenols using response surface methodology (RSM). A central composite response surface design generated from Design-Expert version 7.0.0 (Statease Inc., Minneapolis, MN, USA, Trial version) was used. The selected levels of incubation time (*χ*_1_) and enzyme concentration as U/g (*χ*_2_) for individual enzymes while pH (*χ*_1_) and Temperature (*χ*_2_) for mixed enzymes that result in the highest polyphenol content (data not shown) were 12, 24, 36, and 48 h; 500, 1000, 1500, 2000, and 2500 U/g; 4, 4.5, 5, 5.5, 6, 6.5, 7, and 7.5; and 35, 40, 45, 50, and 60 °C, respectively.

Browne–Forsythe and one-way analysis of variance (ANOVA) test with post hoc Tukey’s test (*p* < 0.05) were used to assess response variance homogeneity and significant effects among various treatments (combinations), respectively.

The response data were fitted to a second-order polynomial equation, which characterized the effect of the independent factors on the response as well as their combined effect on the response Y and determined the interrelationship between the test variables. To fit the experimental data, a reduced cubic model with linear, squared, and interaction factors was used. In the response surface analysis, the generalized second-order polynomial model was utilized, as given by (Equation (1)):Y = a_o_ + a_1_ *χ*_1_ + a_2_ *χ*_2_ + a_12_ *χ*_1_ *χ*_2_ + a_11_ *χ*_1_^2^ + a_22_ *χ*_2_^2^
(1)
where Y (i = 1–5) is the expected total phenolic content (TPC) response, a_o_ the estimated regression coefficient of the fitted response at the central point of the model, a_1_, a_2_, and a_3_ the coefficient of regression for linear effect expressions, a_11_, a_22_, and a_33_ the quadratic effects, and a_12_, a_13_, and a_23_ the effects of interaction.

ANOVA was used to determine the statistical significance of the model and its various terms. In addition to R^2^, adjusted R^2^, and anticipated R^2^ values, the lack of fit test was employed to assess the adequacy of created models.

### 2.5. Enzyme-Assisted Extraction

The enzyme-assisted extraction was done as reported by Fernández et al. [11] with some adjustments. About 1 g corn tassel was extracted in 50 mL of buffer solution (0.1 M sodium acetate or 0.1 M sodium phosphate based on the pH of the used enzyme) containing enzyme under evaluation (cellulase, protease, and their mixture). Firstly, different incubation times (*χ*_1_) (12–60 h) and concentrations (*χ*_2_) (500–2500 U/g) for each enzyme (Cellulase and protease) were used. Secondly, the effect of temperature (*χ*_1_, 40 to 55 °C) and pH (*χ*_2_, 5 to 7) values of the cellulase–protease mixture with ratio (1:1) was also investigated. The hydrolysis conditions were chosen on the basis of the optimum temperature, pH, enzyme concentration, and incubation time for each enzyme as specified on the data sheets provided by the enzyme dealers. The extraction of corn tassel by 60% ethanol–water solution was accomplished in a precise shaking incubator (WiseCube WIS-20, Daihan scientific, Seoul, South Korea) with gentle agitation (200 rpm) in the dark. Finally, the sample was filtered utilizing Whatman No. 4 filter paper and stored at −18 °C in a brown flask till analysis.

### 2.6. Total Phenolic Compounds and HPLC Analysis

Total phenolic compounds of dried corn tassel extract were estimated by Folin–Ciocâlteu method based on Abedelmaksoud et al. [12], and results were expressed as mg Gallic acid equivalents (GAE)/g powder.

The phenolic composition of the optimal corn tassels extract was determined using an Agilent 1260 series HPLC system (Agilent Technologies Inc., Santa Clara, CA, USA) according to Elsayed, Hammad, and Abd El-Salam [10].

### 2.7. Antioxidant Activity

DPPH radical inhibition efficiency of the investigated Corn tassels extract was determined according to Elsayed et al. [13]. The absorbance of the samples was measured using the UV-visible spectrophotometer (Unico UV-2000, Dayton, NJ, USA) at 517 nm. Butylated hydroxytoluene was used as a standard.

ABTS radical scavenging activity was evaluated using the ABTS method [14]. The results are expressed as a percentage of quenched radicals according to the equation: ABTS scavenging activity (%) = (Ac − As)/Ac × 100, where Ac is the absorbance of ABTS, and As is the absorbance of the sample.

### 2.8. Antimicrobial Activity

Tested microorganism, the inhibitory effect of corn tassels extracts was examined against five strains of food-borne pathogenic bacteria. Two Gram-positive bacteria Bacillus cereus EMCC 1080, Staphylococcus aureus ATCC 13565, and three gram-negative bacteria *Salmonella typhi* ATCC 25566, *Escherichia coli* 0157 H7 ATCC 51659, and *Pseudomonas aeruginosa* NRRL B-272. Eight fungal species were used for antifungal assay, containing *Aspergillus flavus* NRR 3357, *A. parasiticus* SSWT 2999, *A. ochraceus* ITAL 14, *A. niger* IM I288550, *A. westerdijikia* CCT 6795, *A. carbonarius* ITAL 204, *Fusarium proliferatum* MPVP 328, and *Penicillium verrucosum* BFE 500.

The sensitivity test of corn tassels extracts was determined with various bacterial and fungal cultures using disc diffusion method by the Kirby–Bauer technique [15,16]. DMSO was a negative control, and ceftriaxone, commercial fungicide (miconazole nitrate, 1 mg mL^−1^), was used as a positive control for bacterial and fungal, then the inoculated plates were incubated at 37 °C for 24 h for bacterial and 25 °C for 24–48 h for fungal. At the end of the incubation period, inhibition zones were measured and expressed as the diameter of the clear zone with the diameter of the paper disc. All treatments consisted of three replicates, and the averages of the experimental results were determined.

Minimum inhibitory concentration (MIC) for corn tassel extracts was determined using the micro broth dilution methods as of Andrews [17]. Two-fold serial dilutions of the different extract concentrations ranging from 5.0 mg mL^−1^ to 10.0 µg mL^−1^ were used. Equal volumes of tested bacteria (10^8^ cfu mL^−1^; 0.5 McFarland standards) were added to each. A 24 h culture of the tested bacterial species was diluted in 10 mL of tryptic soy broth (TSB) with reference to the 0.5 McFarland standard to achieve inoculation of well. MIC values were taken as the lowest concentration of the antimicrobial agent that inhibited bacterial growth after 24 h incubation at 37 °C.

MIC against fungi was performed by using the technique of Marrez and Sultan [18]. Corn tassel extracts at different concentrations were separately dissolved in 0.5 mL of 0.1% Tween 80 (Merck, Darmstadt, Germany), then mixed with 9.5 mL of melting, 45 °C, PDA and poured into a Petri dish (6 cm). The prepared plates were centrally inoculated with 3 µL of fungal suspension (10^8^ cfu mL^−1^; 0.5 McFarland standards). The plates were incubated at 25 °C for 24–48 h. At the end of the incubation period, mycelial growth was monitored, and MIC was determined.

### 2.9. Cytotoxic Activity

The cytotoxic effects of corn tassel extracts by ethanol, cellulose, protease, and mixed enzymatic extraction on human cancerous and normal cell lines were examined by MTT assay according to Senthilraja and Kathiresan [19] against human colorectal carcinoma cell line (Caco-2), human Lung carcinoma cell line (A549), and human lung fibroblast normal cell line (Wi-38).

### 2.10. Statistical ANALYSIS

Chemical assays were performed in triplicate, and the results were represented as mean standard deviations. Statistical analysis was performed using the Originpro 9.6 software (OriginLab Corp., Northampton, MA, USA). One-way ANOVA was applied to analyze the significant differences by GraphPad Prism 5.02 software (GraphPad Prism Software Inc., San Diego, CA, USA). A *p*-value of <0.05 was considered statistically significant.

## 3. Results and Discussion

### 3.1. Optimization of Phenolic Extraction Conditions with the Use of Cellulase, Protease, and Their Mixture (1:1)

The utilization of enzymes for phenolic extraction from corn tassels was conducted using cellulase, protease, and their mixture (1:1). Firstly, different incubation times (*χ*_1_) and enzyme concentrations (*χ*_2_) for each enzyme (cellulase and protease) were used (Appendix A—Figure 1a,b) for RSM to assess their impact on TPC values and optimize extraction parameter. The TPC value in corn tassel extract varied from 5.67 to 9.66 mg/g for cellulase treatments and from 4.67 to 8.69 mg/g for protease treatments (Appendix A). Then, the effects of temperature (*χ*_1_, from 40 to 55 °C) and pH (*χ*_2_, from 5 to 7) values of the enzyme mixture, including cellulose and protease with ratio (1:1) were also investigated (Appendix A—Figure 1c) for RSM to assess the impact of the mixture of enzymes on the TPC values and also optimize extraction parameters. The TPC value of corn tassel extract varied between 6.95 and 10.72 mg/g for the mixture treatment (Appendix A). The influence of cellulase and protease concentrations on the extraction of phenolic compounds from corn tassels was very remarkable for extracting such compounds. It was observed that the extraction amounts of phenolic compounds in corn tassel extracts by enzyme-assisted method increased by 41.62% (cellulase), 25.62% (protease), and 59.11% (cellulase–protease mixture 1:1) compared to that in the control sample (6.75 mg/g) at the respective optimum condition was observed (Table 1).

Because the cell wall is composed of cellulose, hemicellulose, pectin, and protein, as well as phenolic compounds linked to available polysaccharides via hydrogen and hydrophobic bonds, hydrolyzing enzymes such as cellulase, pectinase, and hemicellulose can be used to destroy the cell wall structure. These enzymes can also be utilized to improve the penetrating capacity of cell walls, resulting in the release of phenolic compounds and a higher extraction yield of bioactive substances [20]. Another mechanism is most likely the enzyme’s direct action on the breakdown of ester or ether bonds between phenols and plant cell wall polymers [21]. Furthermore, cellulase impacts the cellulose found beneath the main layer and the midline lamella of plant cells. The main layer is made up of a stiff and strong cellulose skeleton that is embedded in a gel-like matrix of hemicellulose, pectin, and glycoprotein. All responses had a significant sum of squares and higher regression coefficients, indicating compliance with the variables set with a second-order polynomial (see Equations (2)–(4)). To obtain regression models, linear, interactive (2 FI), quadratic, and cubic models were fitted to the experimental data, and the results are shown in Appendix A. From Appendix A, the quadratic model was the best fit for modeling experimental data. Thus, the values of the quadratic model’s coefficient of determination (*R*^2^), modified coefficient of determination (Adj-*R*^2^), and expected coefficient of determination (Pred-*R*^2^) were among the highest and most significant (*p* < 0.001). The multiple regression equations generated in uncoded form between the various responses and process variables are shown below (Equations (2)–(4)):TPC (Cellulase) = 9.57 + 0.76*χ*_1_ + 0.57*χ*_2_ − 0.15*χ*_1_*χ*_2_ − 1.47*χ*_1_^2^ − 0.56*χ*_2_^2^(2)
TPC (Protease) = 7.89 + 1.24*χ*_1_ + 0.066*χ*_2_ − 0.18*χ*_1_*χ*_2_ − 1.50*χ*_1_^2^ + 0.29*χ*_2_^2^(3)
TPC (mix) = 7.87 − 1.67*χ*_1_ − 0.19*χ*_2_ + 0.010*χ*_1_*χ*_2_ + 0.93*χ*_1_^2^ + 0.031*χ*_2_^2^(4)

Appropriate model parameters (*p* < 0.05) are *χ*_1_ (enzyme incubation time), *χ*_2_ (enzyme concentration), *χ*_1_χ_2_, *χ*_1_^2^, *χ*_2_^2^ in the case of cellulase and protease treatments. While in case of mixed enzyme (1:1), *χ*_1_ (Temperature), *χ*_2_ (pH), *χ*_1_*χ*_2_, *χ*_1_^2^, *χ*_2_^2^, are significant model terms (*p* < 0.05). Regression coefficients magnitude designated maximum positive effect of mixed enzyme (TPC was 10.74 mg/g) followed by cellulase (TPC was 9.56 mg/g) and protease (TPC was 8.48 mg/g) (Table 1).

Figure 1a,b illustrated the TPC increased significantly with increasing incubation time (factor A) till 36 h and then decreased subsequently. It was also found that there was no significant effect for factor B (enzyme concentration) on the TPC in corn tassel extract by protease-assisted extraction (Figure 1b). Therefore, surface plots of the model equation, disturbance, and 3D response demonstrated that incubation time has a relatively bigger effect on TPC values for cellulase- or protease-assisted extraction than enzyme concentration. For enzyme-assisted extraction with the combined use of cellulase and protease, the effect of incubation temperature on the TPC in corn tassel extract was bigger than that of pH. The optimal combination of factors with the highest desirability value was chosen to be 1.00, 0.95, and 0.99 for cellulase, protease, and their mixture, respectively. The optimum conditions are presented in Table 1. Corn tassel extraction by solvent and enzyme-assisted method was conducted under these conditions in the lab, and the experimental response values matched the anticipated value using the model obtained for the optimized corn tassel to extract TPC. Hence, the fitted models were determined to be suitable for predicting the responses (Table 1).

### 3.2. Identification of the Phenolic Compounds in Corn Tassel Extract Obtained by Enzyme-Assisted Method

HPLC was used to identify the phenolic compounds in corn tassel extracts and evaluate for the first time the effect of the enzyme used on the composition of phenolic compounds. From Appendix A and Table 2, the extraction technique has a slight influence on the varieties of phenolic components; 14 compounds were detected in corn tassel extract by cellulase–protease-mixture-assisted method and 12 compounds in the extract by cellulose- or protease-assisted method. Ellagic acid and taxifolin were not detected in the extract by cellulose-assisted extraction, while kaempferol and coumaric acid were not detected in the extract by protease-assisted extraction. The common phenolic compounds in corn tassel extracts obtained by three methods were listed as gallic acid, chlorogenic acid, caffeic acid, naringenin, rutin, ferulic acid, syringic acid, and catechin, and their contents except catechin and pyrocatechol for the sample from mixed-enzyme-assisted extraction were significantly higher than that for the sample from single-enzyme-assisted extraction (Table 2). The finding indicated that mixed-enzyme-assisted extraction could lead to an increase in the number of phenolic compounds in corn tassel extracts compared to cellulase/protease-assisted extraction. Cerda et al. [22] reported that the extraction by enzymes raises the number of phenolic compounds extracted in addition to the antioxidant activity, the mechanism of enzymatic extraction is the degradation of the network of polysaccharides in the plant cell wall where this network retains the phenolic compounds. Another mechanism of enzymatic extraction is direct catalysis causing distraction of the ether and/or ester bonds between the phenols and polymers of the plant cell wall [21].

### 3.3. Antioxidant Activity of Corn Tassel Extracts

The assays of DPPH and ABTS radical scavenging were used to evaluate the antioxidant capacity of corn tassel extracts from different extraction methods, and the results are shown in Table 3. The results presented significant variation in DPPH radical scavenging activity for the extracts from different extraction methods. The DPPH and ABTS radical scavenging activities were observed to be significantly higher in the corn tassel extracted by enzymes (83.20–92.98% and 88.70–95.18%, respectively) compared to ethanol extract (75.25% and 80.25%, respectively) and also BHT (87.69% and 83.74%, respectively). The extract obtained by mixed enzyme showed the highest antioxidant activity, followed by cellulose > protease > ethanol. Our results were similar to Cerda, Martínez, Soto, Poirrier, Perez-Correa, Vergara-Salinas, and Zúñiga [22], who reported that enzyme-assisted extraction raised the number of phenolic compounds in the extracts, also resulting in an increase in the antioxidant activity compared to the ethanolic extraction. de Vries and Visser [23] attributed the efficiency of extraction with enzymes to an increase in the antioxidant activity due to 90% of the plant cell wall being formed from polysaccharides, which consist of cellulose, hemicellulose, and pectin. Enzymatic analysis of the plant wall polysaccharides with certain enzymes led to a rise in the release of compounds combined with the cell wall components, and this is a good replacement for conventional extraction methods; this explains the reasons for the superiority of cellulase over protease in antioxidant activity. Additionally, Shahidi [24] found that the radical scavenging capacity of protease extracts could result from their ability to enhance the extraction capacity of phenolic compounds which was known as strong DPPH radical scavengers.

### 3.4. Antimicrobial Activity of Corn Tassel Extracts

The antibacterial activity of corn tassel extracts by ethanol, cellulose, protease, and their binary-mixture-assisted extraction against two Gram-positive and three Gram-negative bacteria of food borne pathogenic are shown in Appendix A. The corn tassel extract from enzyme-mixture-assisted extraction had the highest inhibition zone (12.8, 11.5, 10.8, 10.5, and 10.2 mm, respectively) against all borne pathogenic bacteria (*E. coli*, *P. aeruginosa*, *S. typhi*, *B. cereus*, and *S. aureus*) compared to other extracts. In addition, the corn tassel extracts by ethanol had a relatively lower inhibition zone for all pathogens bacterial, with a diameter of the inhibition zones varying from 5.7 to 7.9 mm. The results also showed that the corn tassels extract by protease exhibited moderated inhibitory activity against *B. cereus* (8.7 mm) and *S. aureus* (8.8 mm) compared to the positive control. As shown in Figure 2a, the minimum inhibitory concentrations (MIC) of corn tassel extract from enzyme-mixture-assisted extraction against *tested bacterial strains* was observed to be 0.3–0.7 mg/mL, which were significantly lower than that of the extracts by cellulase- or protease-assisted extraction (1.3–2.8 and 1.6–3.2 mg/mL, respectively) and also ethanol extraction (1.8–4.9 mg/mL). The finding suggested that the antibacterial activity of corn tassel extract by mixed-enzyme-assisted extraction was stronger than that of the extracts by single-enzyme-assisted extraction and ethanol extraction.

Additionally, the inhibition zone diameter of corn tassel extracts against mycotoxigenic fungus is shown in Appendix A. The corn tassel extracts had antifungal activity against all tested mycotoxigenic fungi. Meanwhile, the corn tassel extract by a cellulase–protease mixture showed the strongest antifungal activity, 11.0 mm against *A. westerdijikia*, followed by *A. niger* and *P. verrucosum* with inhibition zone 10.7 and 10.0 mm, respectively. The corn tassel extract by ethanol showed the lowest antifungal activity against all examined fungi, and the highest inhibition zones, 7.2 and 7.1 mm, were observed against *A. carbonarius* and *A. westerdijikia*, respectively, while the lowest activity was observed against *A. flavus* with inhibition zone 6.1 mm.

Minimum inhibitory concentrations of corn tassel extracts against mycotoxigenic fungi are shown in Figure 2b. The lowest MIC value was observed in corn tassel extracts by mixed-enzyme-assisted method, followed by corn tassel extracts by cellulase, protease, and ethanol extraction. The highest activity in corn tassel extracts by mixed-enzymes-assisted extraction and corn tassel extracts by cellulase were observed against *A. niger* and *P. verrucosum* with MIC values of 0.5 and 0.7 mg/mL, respectively; however, the corn tassel extract by ethanol and cellulase recorded the lowest activity against *A. niger,* with MIC values 4.9 and 4.00 mg/mL, respectively. The antimicrobial activity of corn tassels extracts could be attributed to the high phenolic compounds such as gallic acid, chlorogenic acid, and caffeic acid, which had higher contents in the extract by cellulase–protease-mixture-assisted extraction than that by other extraction methods (Table 2). The phenolic compounds could damage the bacterial cell by decaying the cell wall, stopping the turn of ion channels, and inhibiting the synthesis of adenosine triphosphate [25]. Accordingly, the antimicrobial activity of corn tassel extract by mixed-enzyme-assisted extraction was stronger than that of the extracts by single-enzyme-assisted extraction and ethanol extraction.

### 3.5. Cytotoxic Activity of Corn Tassel Extracts

Cytotoxicity was expressed as the percentage growth inhibition of Caco-2, A549, and Wi-38 cells treated with various doses of extracts (from 1000 to 31.25 μg/mL). Doxorubicin, an anticancer drug for hematologic and solid tumors, was used as the positive control. From Figure 3, some of the extracts caused cell cytotoxicity in a concentration-dependent manner. Probit analysis was carried out for IC_50_ determination, and the results are shown in Table 4. The mixed enzyme extract had the highest inhibitory activity against the Caco-2 cell line (IC_50_, 392.62 μg/mL) and A549 cell line (IC_50_, 210.66 μg/mL) compared to the extracts obtained by free enzyme and ethanol extraction, and a low toxicity effect on the Wi-38 cell line (IC_50_, 809.85 μg/mL). The observation indicated that corn tassel extract obtained by mixed-enzyme-assisted extraction was nontoxic to normal lung fibroblast cell line but cytotoxic to human colorectal and lung cancer cell lines with the dose-dependent effect. The antiproliferative activity of doxorubicin was found to be more potent against the Caco-2 cell line (IC_50_, 110.83 μg/mL) and A549 cell line (IC_50_, 61.42 μg/mL) and had the highest toxicity effect on the Wi-38 cell line (IC_50_, 122.72 μg/mL).

Chemotherapy is one of the most efficient cancer therapies, but its efficiency is limited due to significant drug-related side effects. Extended chemotherapy treatment reduces the immune system response, making patients more susceptible to various diseases and infections. Doxorubicin is a powerful anticancer medication that can be used to treat both hematologic and solid cancers. However, it has the potential to produce multi-organ toxicity in a variety of patients. Doxorubicin toxicity can be attributed to apoptosis, inflammation, and oxidative stress, among other mechanisms [26]. Plants have been reported to treat a variety of cancers and they provide leads for the creation of possible new drugs [27,28]. The most widely dispersed natural minor metabolites in plants are phenolic compounds. Flavonoids are a significant category of phenolics that play a key role in functional extracts [29,30]. Corn tassels are a valuable by-product and a valuable source of bioactive phytochemicals [31].

The variations observed among corn tassel extraction methods were found in HPLC and antioxidant activities results as mentioned above. The results showed that the effect of the extraction method was the main reason for the difference in the cytotoxicity. Corn tassel extract by enzyme-mixture-assisted method recorded the highest phenolic content and antioxidant capacity. Therefore, in previous cytotoxic screenings with different extracts, the score of cytotoxic responses of cellulase–protease mixture extract was higher than the ethanol, cellulase and protease extracts against Caco-2 and A549 cell lines. Phenolic compounds, which are isolated from dried corn tassel, are bioactive phytochemicals with antioxidant properties [3]. Additionally, the corn tassel consists of polysaccharide, saponin, and flavonoids, which can inhibit the proliferation of MGC80-3 gastric cancer cells in vitro. Furthermore, “Tasselin A”, a pure 4-hydroxyl-1-oxindole-3-acetic acid isolated from sweet corn tassels, was found to have a strong antioxidant potential [32,33]. Corn bioactive compounds also decreased levels of Ras protein and nitric oxide synthases (NOS) inflammatory enzymes and decreased mRNA expression of cytokines such as tumor necrosis factor-alpha (TNF-α) and interleukin 6 (IL-6) while increasing the free radical detoxifying enzymes mRNA expression. Corn extracts activated caspase-3 and lowered the levels of Ras protein in tumor cells at the molecular level [34,35,36,37].

## 4. Conclusions

The recovery of phenolic compounds from corn tassel was well improved by enzyme-assisted extraction compared to ethanolic extraction. Especially TPC in corn tassel extract was greatly enhanced by the combined use of cellulase and protease (1:1). Under the optimal enzyme pretreatment conditions obtained by RSM, the TPCs in corn tassel extracts by cellulase-, protease-, and their mixture-assisted extraction were up to 9.56, 8.45, and 10.70 mg/g, respectively. Furthermore, more varieties of phenolic compounds in corn tassel extract by mixed-enzyme-assisted extraction were quantitatively identified in corn tassel extract by mixed-enzyme-assisted extraction than that by single-enzyme-assisted extraction. Additionally, based on the DPPH and ABTS free radical scavenging assay, the antioxidant activity of corn tassel extract by mixed-enzyme-assisted extraction was significantly stronger than that of the single-enzyme-assisted extracts, ethanolic extract, and also BHT (*p* < 0.05). Corn tassel extract by mixed-enzyme-assisted extraction showed remarkably better antimicrobial properties against all tested food-borne pathogenic bacteria and mycotoxigenic fungi than CTE by other extraction methods. It was also confirmed that corn tassel extract obtained by mixed-enzyme-assisted extraction was nontoxic to normal lung fibroblast cell line (Wi-38) but cytotoxic to human colorectal and lung cancer cell lines (Caco-2 and A549) with the dose-dependent effect.

## Figures and Tables

**Figure 1 foods-11-02145-f001:**
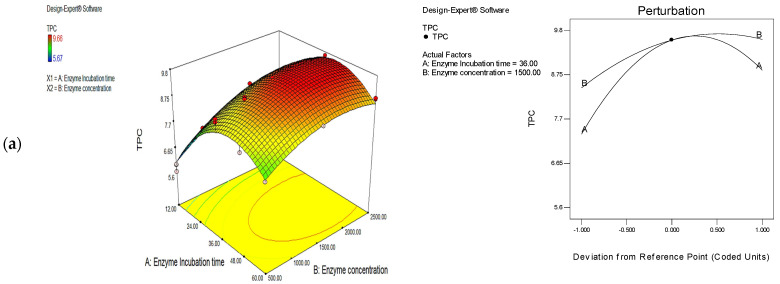
The response surface plots and perturbation plots of the effects of incubation time, enzyme concentration, incubation temperature, and pH on the TPC values in corn tassel extract by cellulase (**a**), protease (**b**), and their mixture (**c**) assisted extraction.

**Figure 2 foods-11-02145-f002:**
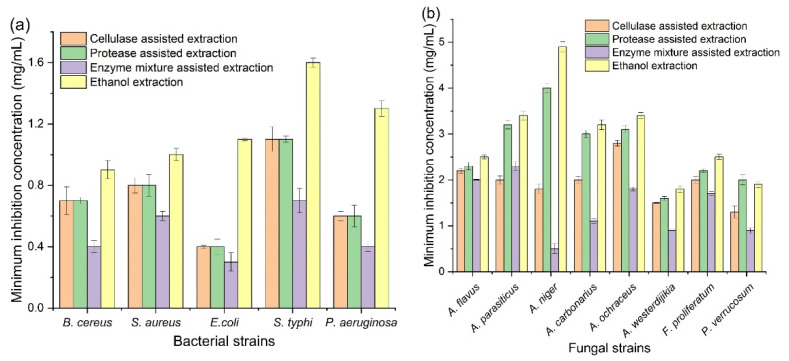
Minimum inhibitory concentrations of corn tassel extracts by different extraction methods against food-borne pathogenic bacteria (**a**) and mycotoxigenic fungi (**b**).

**Figure 3 foods-11-02145-f003:**
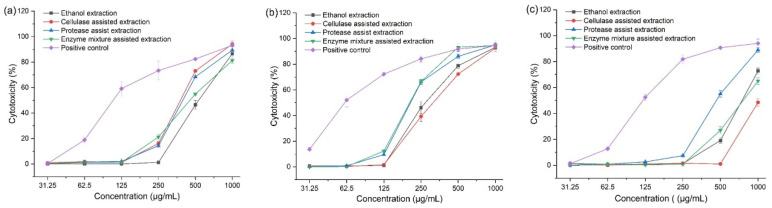
Cytotoxicity of corn tassel extracts by different extraction methods in Caco-2 (**a**), A549 (**b**), and Wi-38 (**c**) cell lines. Positive control—doxorubicin.

**Table 1 foods-11-02145-t001:** Predicted and actual values of TPC in corn tassel extracts by cellulase-, protease-, and their mixture (1:1)-assisted extraction.

Treatment	Enzyme Incubation Time (h)	Enzyme Concentration (U/g)	TPC (mg/g)		Desirability
			Predicted Value	Actual Value *	
Cellulase	39.32	1978.60	9.78	9.56 ± 0.11 ^b^	1.000
Protease	47.38	500.00	8.45	8.48 ± 0.12 ^c^	0.941
Enzyme mixture (40 °C, pH 5.0)	-	-	10.70	10.74 ± 0.10 ^a^	0.995
Ethanol 60%	-	-	-	6.75 ± 0.11 ^d^	-

* Values are expressed as mean ± standard deviation (*n* = 3), and different letters represent a significant difference at *p* < 0.05. Exzyme mixture—cellulase and protease (1:1). -: not included values according to RSM outputs.

**Table 2 foods-11-02145-t002:** Phenolic compounds in corn tassel extract by enzyme-assisted method.

Phenolic Compounds	Cellulase (µg/g)	Proteases (µg/g)	Cellulase–Protease Mixture (µg/g)
Gallic acid	967.88 ± 34.00	314.87 ± 8.74	1694.36 ± 54.75
Chlorogenic acid	659.8 ± 41.56	97.83 ± 2.09	1172 ± 86.38
Caffeic acid	370.48 ± 20.46	9.75 ± 0.63	661.81 ± 32.40
Naringenin	259.05 ± 20.55	118.76 ± 2.24	270.01 ± 28.44
Rutin	181.24 ± 7.21	159.92 ± 1.18	211.24 ± 11.05
Ferulic acid	26.61 ± 2.87	20.39 ± 0.91	180.39 ± 4.93
Syringic acid	148.18 ± 6.09	66.08 ± 1.09	167.22 ± 5.13
Catechin	198.41 ± 16.96	114.62 ± 3.66	134.23 ± 4.51
Ellagic acid	nd	6.99 ± 0.29	74.95 ± 4.50
Methyl gallate	26.18 ± 3.87	8.77 ± 0.80	58.82 ± 5.39
Kaempferol	6.31 ± 0.47	nd	40.86 ± 3.55
Coumaric acid	36.91 ± 1.68	nd	36.83 ± 0.50
Pyrocatechol	21.30 ± 1.54	11.86 ± 0.82	18.85 ± 0.50
Taxifolin	nd	6.11 ± 0.29	13.80 ± 0.83

Values are expressed as mean ± standard deviation (*n* = 3). nd: not detected.

**Table 3 foods-11-02145-t003:** In vitro antioxidant activity of the different corn tassel extracts from ethanol and enzyme-assisted extraction.

Corn Tassel Extract	Antioxidant Activity (%)	
	DPPH Assay	ABTS Assay
Ethanol extract	75.25 ± 0.10 ^e^	80.25 ± 0.16 ^e^
Cellulose-assisted extract	86.11 ± 0.12 ^c^	90.29 ± 0.14 ^b^
Protease-assisted extract	83.20 ± 0.13 ^d^	88.70 ± 0.18 ^c^
Mixed-enzyme-assisted extract	92.98 ± 0.09 ^a^	95.18 ± 0.15 ^a^
BHT	87.69 ± 0.14 ^b^	83.74 ± 0.17 ^d^

Values are expressed as mean ± standard deviation (*n* = 3). Different letters within each column represent a significant difference at *p* < 0.05. Mixed enzyme—cellulase and protease (1:1).

**Table 4 foods-11-02145-t004:** Cytotoxic effects of corn tassel extracts by different extraction methods against Caco2, A549, and Wi-38 cell lines.

Treatment	IC_50_ (µg/mL)		
	Caco2	A549	Wi-38
Ethanol extraction	624.57 ± 17.66 ^a^	332.14 ± 11.73 ^a^	778.66 ± 61.38 ^b^
Cellulase-assisted extraction	410.51 ± 15.41^c^	359.56 ± 18.11 ^a^	1073.50 ± 49.16 ^a^
Protease-assisted extraction	461.98 ± 18.87 ^b^	213.45 ± 13.66 ^b^	578.89 ± 17.77 ^c^
Mixed-enzyme-assisted extract	392.62 ± 24.3 ^c^	210.66 ± 8.9 ^b^	809.85 ± 37.25 ^b^
Doxorubicin	110.83 ± 6.82 ^d^	61.42 ± 5.79 ^c^	122.72 ± 7.8 ^d^

Values are expressed as mean ± standard deviation (*n* = 3), and different letters whintin each column represent a significant difference at *p* < 0.05. IC_50_ is the concentration at which 50% of cell death occurs. Mixed enzyme—cellulase and protease (1:1).

## Data Availability

Data are contained within the article or Appendix A.

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
