# Peer review of "Phenolic Profiling and In-Vitro Bioactivities of Corn (Zea mays L.) Tassel Extracts by Combining Enzyme-Assisted Extraction"

_foods, 2022, doi:10.3390/foods11142145_

Round 1

Reviewer 1 Report

The manuscript entitled "Phenolic Profiling and In-vitro Bioactivities of Corn (Zea mays L.) Tassel extracts by Combining Enzyme-assisted Extraction" is well designed, executed and written. The phenolic contents in the enzymatically digested extracts of corn tassel was evaluated by HPLC-DAD technique and then evaluated for their antioxidant, antimicrobial and anticancer activity. The provided data supports the findings.

The phenolic contents in the enzymatically (cellulase, protease and their mixture) digested extracts of corn tassel was evaluated by HPLC-DAD technique and then evaluated for their antioxidant (DPPH and ABTS radical scavenging assay), antimicrobial (antibacterial and antifungal) and anticancer activity (MTT assay). The manuscript reported high content of 14 phenolic compounds after enzymatic digestion and by HPLC-DAD technique and reported their potential activities. The manuscript is novel in its content and interesting for the research community.

The study reported for the first time that enzymatic digestion of corn tassel has higher yields of phenolic compounds and potential application in the food industry.Although the phenolic contents in the corn tassel extract has been reported previously, but this study highlighted that enzymatic digestion has higher yields as compared to the previous conventional techniques.

The manuscript is very well written and compiled. The content flow is smooth and easy to read. The experiments were executed in a well-designed manner with experimental controls. The aim and objectives in the study is well addressed.

The manuscript can be  following minor revision.

1.     Please check the manuscript thoroughly for typos and grammatical errors.

2.  Authors should include the HPLC chromatogram of the extracts showing the retention value for each phenolic compound in the manuscript or supplementary material.

Author Response

Response to Reviewer 1 Comments

Point 1: Please check the manuscript thoroughly for typos and grammatical errors.

Response: Thank you for your good suggestion. We have checked the whole manuscript for typos and grammatical errors and revised them in the revised manuscript.

Point 2: Authors should include the HPLC chromatogram of the extracts showing the retention value for each phenolic compound in the manuscript or supplementary material.

Response: Thank you for your good suggestion. We have added the HPLC chromatograms of the extracts as shown in Figure S1 in the supplementary material.

Reviewer 2 Report

The work is very interesting and important for a certain area of research that deals with development aimed at obtaining new materials being the source of bioactive ingredients. Measurements seem appropriate and the description of the experimental methods used is accurate. The manuscript is clearly written although there are editorial errors in some place authors used capitol letter – I do not why. The data are  interpreted correctly but Figure 1 is very small, someone have to use a magnifying glass to see something.

 At the end of Introduction authors wrote that "The results ...would provide a solid foundation for the development of new value-added products from corn tassel" - it is worth mentioning which ones exactly.

At the paragraph 2.2 you should add information on how long the corn tassels were dried and how the consistent weight was determined.

At the paragraph 2.3 you should  add more information about extraction because this is very important step of this work.

I would like to emphasize the great value of this work, as the authors managed to obtain very good adjustments to the models. 

Author Response

Response to Reviewer 2 Comments

Point 1: The manuscript is clearly written although there are editorial errors in some place authors used capital letter – I do not why. 

Response: Thank you for your good comment. We have checked the capital letter used in our manuscript and revised them in the revised manuscript. When  the abbreviations with the capital letter were used for the first time, they were defined well.

Point 2: The data are  interpreted correctly but Figure 1 is very small, someone have to use a magnifying glass to see something.

Response: Thank you for good comment. We have revised Figure 1 to ensure the clear observation in the revised manuscript.

Point 3: At the end of Introduction authors wrote that "The results ...would provide a solid foundation for the development of new value-added products from corn tassel" - it is worth mentioning which ones exactly.

Response: Thank you for your good suggestion. We have revised this sentence in the revised manuscript according to your suggestion.

Point 4: At the paragraph 2.2 you should add information on how long the corn tassels were dried and how the consistent weight was determined.

Response: Thank you for your good suggestion. We have added the information in the revised manuscript.

Point 5: At the paragraph 2.3 you should  add more information about extraction because this is very important step of this work.

Response: Thank you for your good suggestion. We have added more information about extraction in Section 2.3 in the revised manuscript.

Point 6: I would like to emphasize the great value of this work, as the authors managed to obtain very good adjustments to the models. 

Response: Thank you for your positive comment. We are very appreciative for your valuable comments and suggestions.